# Individualised and non-contact post-mortem interval determination of human bodies using visible and thermal 3D imaging

Leah S. Wilk [1,2], Gerda J. Edelman[3], Martin Roos[3], Mara Clerkx[4], Inge Dijkman[4], Jordi Vera Melgar[4], Roelof-Jan Oostra [2,4] & Maurice C. G. Aalders [1,2✉]

Determining the time since death, i.e., post-mortem interval (PMI), often plays a key role in forensic investigations. The current standard PMI-estimation method empirically correlates rectal temperatures and PMIs, frequently necessitating subjective correction factors. To overcome this, we previously developed a thermodynamic finite-difference (TFD) algorithm, providing a rigorous method to simulate post-mortem temperatures of bodies assuming a straight posture. However, in forensic practice, bodies are often found in non-straight postures, potentially limiting applicability of this algorithm in these cases. Here, we develop an individualised approach, enabling PMI reconstruction for bodies in arbitrary postures, by combining photogrammetry and TFD modelling. Utilising thermal photogrammetry, this approach also represents the first non-contact method for PMI reconstruction. The performed lab and crime scene validations reveal PMI reconstruction accuracies of 0.26 h ± 1.38 h for true PMIs between 2 h and 35 h and total procedural durations of ~15 min. Together, these findings broaden the potential applicability of TFD-based PMI reconstruction.

[1] Department of Biomedical Engineering and Physics, Amsterdam UMC Location AMC, University of Amsterdam, Meibergdreef 9, 1105AZ Amsterdam, The Netherlands. [2] Co van Ledden Hulsebosch Center, University of Amsterdam, Science Park 904, 1098XH Amsterdam, The Netherlands. [3] Netherlands Forensic Institute, Divisie Bijzondere Dienstverlening en Expertise, Laan van Ypenburg 6, 2497 GB The Hague, The Netherlands. [4] Department of Medical Biology, Section Clinical Anatomy and Embryology, Amsterdam UMC Location AMC, University of Amsterdam, Meibergdreef 9, 1105 AZ Amsterdam, The Netherlands. ✉email: m.c.aalders@amsterdamumc.nl

Knowing the time since death, i.e., the post-mortem interval (PMI), is essential in reconstructing the timeline of events preceding death and therefore plays a crucial role in forensic investigations. As a result, devising methods to determine the post-mortem interval continues to be an integral challenge in forensic medicine[1]. In this context, various strategies have been explored to measure the PMI based on post-mortem pathophysiological changes. These strategies generally belong to one of two groups. The first seeks to quantify thanatochemical changes such as nucleic acid degradation[2–4], changes in the ocular potassium concentration[5–7], as well as microbial[8] and metabolomic[9] changes. These necessitate physical sampling of human tissue or bodily fluids[10] followed by further examination in a laboratory. Conversely, the second group consists of approaches that involve quantification of optical, mechanical or thermal changes in human tissue[11–15]. These approaches, therefore, do not require sample extraction or laboratory examinations, and hence potentially allow application in situ, i.e., directly at the crime scene.

Of the latter, the change in body temperature is one of the most widely studied, with its precedence dating back as far as the 19th century[16]. Consequently, many models have been conceived to correlate the post-mortem change in body temperature with the PMI[16–20]. One such model[21–23], developed by Claus Henssge in the late 1970s, is the current standard method for PMI determination in forensic practice. Usually presented in the form of a nomogram (and therefore referred to as Henssge's nomogram), this model utilises the rectal (core) temperature to determine the PMI. The underlying relation between the core temperature and the PMI was determined empirically on the basis of a data set of body weights, coverages and surface contacts. As a result, this approach is subject to several considerable limitations[24]. First, deviations from the circumstances under which the data underpinning the model were recorded necessitate the use of subjective correction factors or preclude its use altogether. Second, as this model uses core temperatures to reconstruct PMIs, it intrinsically relies on rectal thermometry, increasing the risk of trace contamination and destruction. Third, the model classifies bodies only by weight, omitting the fact that the core temperatures of two bodies, equal in weight but differing in stature, posture or internal composition, will change at different rates. Consequently, PMIs estimated using Henssge's nomogram are subject to relatively large uncertainties, from ±2.8 h at best and up to ±7 h.

The technological and computational strides made since the conception of Henssge's nomogram now enable researchers to address these limitations. Most notably, several numerical approaches which model post-mortem body cooling have been developed in recent years[25,26]. However, these approaches are yet to be validated experimentally on deceased human bodies. Therefore, we recently developed a numerical approach for thermometric PMI reconstruction that combines a comprehensive thermodynamic finite-difference (TFD) model and thermometry of the skin, which we validated on deceased human bodies[27]. In this approach, manually determined anthropometric data (lengths, diameters and circumferences of various body parts) are used to render a computational model of the body, consisting of 3D geometric shapes, assuming a straight posture. By using this 3D model in conjunction with the laws of thermodynamics and the relevant thermal parameters (e.g., the ambient temperature and the thermal properties of the body and the surrounding materials etc.), our TFD model accurately simulates the heat exchange between the body and its environment, yielding spatially-resolved simulations of the body temperature as a function of the PMI. The time point at which the simulated temperature best approximates the measured temperature (at the same body location) then corresponds to the reconstructed PMI. While this approach achieves highly accurate PMI reconstructions, the fact that the model assumes a straight body configuration may limit its applicability for bodies found in other (i.e., non-straight) postures.

In this work, we develop and validate an integrated computational framework to overcome this limitation, by combining computer vision and TFD modelling. To this end, we employ Structure-from-Motion (SfM)[28–35] to reconstruct the exact shape and posture of individual bodies from a series of RGB images. Using this information, we develop an individualised TFD model, allowing simulation of the temperature of bodies in arbitrary postures as a function of the PMI. Moreover, by exploiting thermal SfM to measure the corresponding 3D temperature distribution of those bodies, our approach also represents the first non-contact method for PMI determination. We evaluate the performance of this photogrammetry-based approach in a morgue setting for six recently deceased human bodies at post-mortem intervals ranging from 2 h to 35 h. This reveals the lowest errors for PMI reconstruction to date, with average accuracies as high as 0.26 h ± 1.38 h (16 min ± 83 min), even at progressed cooling stages. Furthermore, we carry out a field validation study in which we test our approach at four real crime scenes in both indoor and outdoor settings. Here, a timeframe for the time since death is known through circumstantial information. Moreover, using rectal temperature data from these cases, we also benchmark our approach against the current standard method (Henssge's nomogram). TFD-based predictions all fall within the circumstantial timeframes, as well as within the timeframes predicted by the standard method, while achieving a narrower timeframe for the time since death.

## Results

**Experimental design**. We first established two orthogonal approaches to obtain co-registered anthropometric and thermometric data of deceased human bodies, which then served as modelling inputs for TFD-based PMI reconstructions. This is detailed in Fig. 1 and Methods. The first approach consists of using temperature sensors (loggers) (Fig. 1a) to record skin temperatures and a standard digital camera to acquire RGB images of the body. The second approach uses a thermal camera (Fig. 1a) to simultaneously record RGB and thermal images of the body. In both procedures, spatial co-registration of the body shape and posture (anthropometric modelling input) with the skin temperature (thermometric modelling input) is achieved by means of coded imaging targets. The latter were placed at five distinct measurement locations on the body: the forehead, the chest, the abdomen, the thigh and the upper arm (Fig. 1b). In order to generate scaled 3D models of all bodies, partly overlapping RGB images were recorded under ambient lighting conditions at three distinct heights at several locations around the bodies (Fig. 1c). Using SfM, points on the body surface, visible from at least two different viewing angles, were then reconstructed geometrically.

Next, we individualised our TFD model by first generating and then translating triangle surface meshes of the scaled 3D models into volumetric representations of the bodies within 3D cubic grids (Fig. 1d). This step is essential for individualised TFD modelling, as our algorithm relies on a cubic discretisation of the spatial domain. Importantly, these volumetric representations inherently still include the dimensions and posture of the body. The 3D coordinates of the locations of the (automatically detected) coded imaging targets were registered within the surface mesh and then also converted to a location within the 3D cubic grid (Fig. 1d). This co-registration step allows comparison of the skin temperatures measured at those locations (using either the temperature loggers or the thermal camera) with

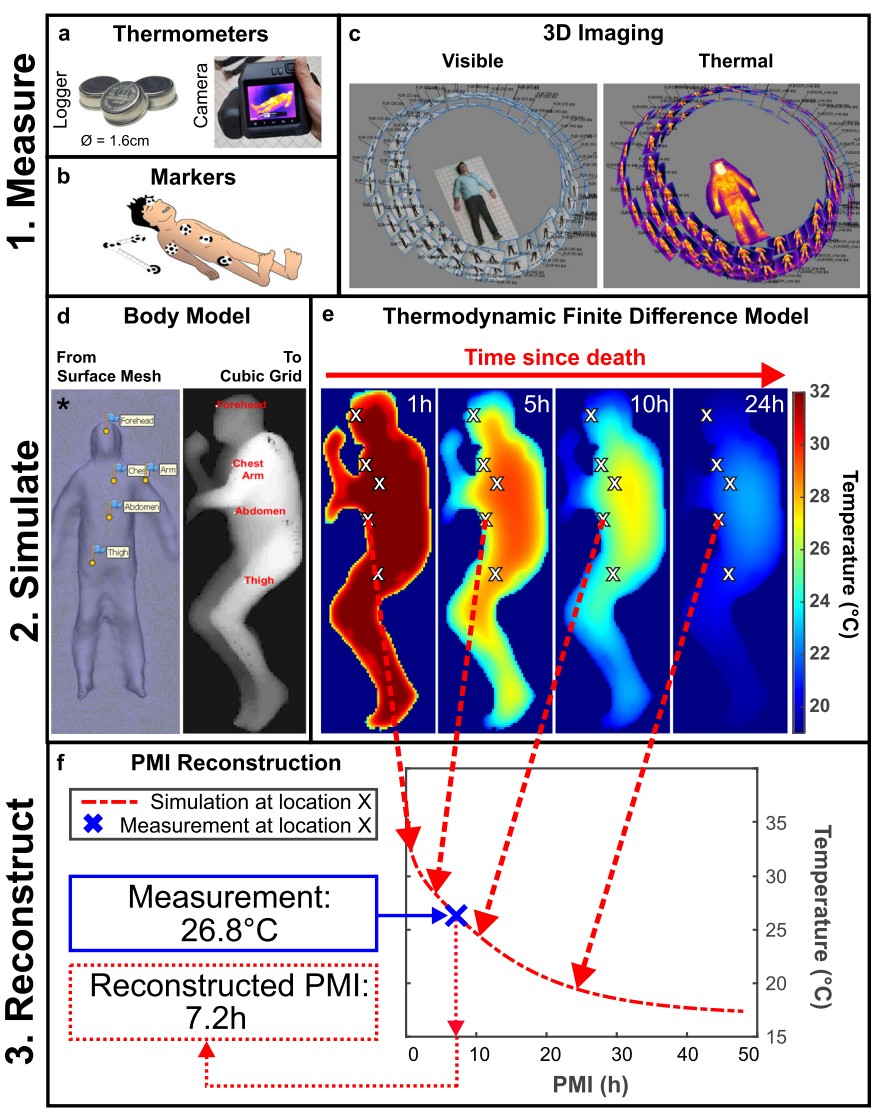

**Fig. 1 Method workflow. a** The two types of thermometers used in this study: thermal loggers and a thermal camera. **b** The coded imaging targets (markers) used to scale the 3D model (attached to the ruler/scale bar next to the body) and co-register simulated and measured skin temperatures (attached to the loggers that are placed on the body). **c** Example of camera positions for visible and thermal photogrammetry. **d** Translation of a surface mesh generated by visible photogrammetry (including automatically detected marker positions) into a cubic grid used for the individualised TFD simulations. **e** Spatially-resolved simulations of the body temperature of the body on the right in **d** at 1 h, 5 h, 10 h, and 24 h after death, computed using the individualised TFD model. **f** PMI reconstruction process: by recording the simulated skin temperature at a specific body location (e.g., as indicated by the red dashed arrows) for many consecutive times after death, we effectively generate a look-up table collating the location-specific skin temperature as a function of the PMI. The temperature value within this look-up table which best approximates the co-registered measured skin temperature then yields the reconstructed PMI. (*) Note that images marked with an asterisk (in **a**, **c**, and **d**) contain representative images (serving as examples of the method) collected from a living subject, in order to protect the privacy of the deceased subject shown in the images without asterisk (in **d** and **e**).

their simulated counterparts at the corresponding position in the cubic grid (Fig. 1e). These spatially co-registered simulations of the body temperature as a function of the PMI were computed using our individualised TFD model. The time point at which the simulated skin temperature at a specific body location (which corresponds to a single cube in the individualised TFD model) best approximates its spatially co-registered measured skin temperature then corresponds to the reconstructed PMI (Fig. 1f).

**Benchmarking of the thermometers and the scaled 3D models**. First, we compared the two types of thermometers used in this study (temperature loggers and thermal camera) with each other. To this end, we extracted 155 skin temperatures from thermal images and compared them to their corresponding co-registered

temperature logger data. In this experiment, perfectly congruent measurement locations are impossible, as the temperature loggers obscure the relevant tissue in the thermal images. Therefore, the pixels around the border of the loggers were selected. These border pixels were identified in the thermal images by placing circles around the loggers, yielding a set of thermal pixels per logger (Fig. 2). Using these values, we computed the average skin temperature around the loggers. Furthermore, in order to ensure comparison of concurrent logger and camera measurements, we utilised simultaneously recorded thermal images and logger data. The resulting data are shown in Fig. 3a and exhibit a strong correlation (linear correlation coefficient: 0.991). We found an average difference of $-0.27 \pm 0.61\,°C$ between the two thermometers.

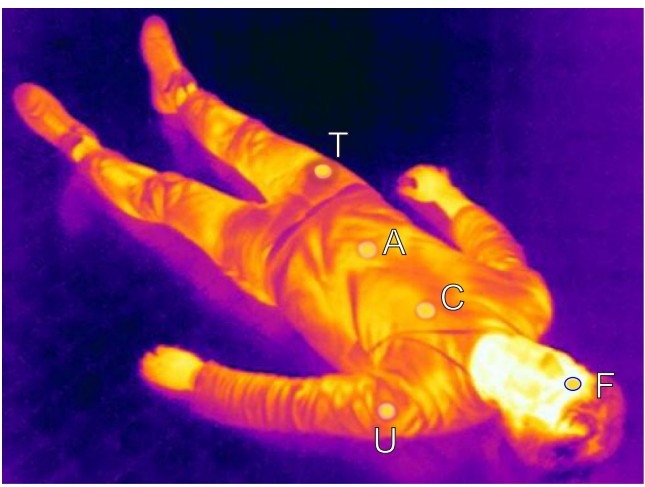

**Fig. 2 Extraction of co-registered thermal imaging data.** Thermal image of a human body, showing circular imaging targets marking the location of the temperature loggers on the forehead (F), the chest (C), the abdomen (A), the upper arm (U) and the thigh (T). The blue circle around the target on the forehead denotes the pixels used to extract the thermal imaging data at this measurement location. Note that this is a representative image (which serves as an example of the method) collected from a living subject, in order to protect the privacy of the deceased subjects of our study.

In order to validate the geometric accuracy of all scaled 3D models, manual reference measurements were compared to their corresponding values in the scaled 3D model. To this end, both arms, both legs and both feet were marked with two coded imaging targets each. Additionally, a reference scale bar marked with three coded imaging targets was placed next to the feet. All of these targets were recognised and labelled automatically in the 3D model of the body. The pairwise distance between these targets was measured both manually, using a tape measure, and virtually, by placing digital scale bars between the labelled targets in the 3D model. These measured distances are compared in Fig. 3b and also exhibit a strong correlation (linear correlation coefficient: 0.993). We found the average difference for the body measurements and the scale bars to be −0.3 cm ± 0.6 cm and 0.1 cm ± 0.2 cm, respectively.

**PMI reconstructions using visible 3D imaging and temperature loggers.** Using the first approach (i.e., temperature loggers and visible photogrammetry), we reconstructed PMIs based on the scaled 3D models and co-registered temperature logger data of six bodies at five body locations. Panels a–e in Fig. 4 show the reconstructed PMIs against the corresponding true PMIs, for each of the five measurement locations (logger-based data are shown as circles). To assess the accuracy of these reconstructed PMIs, we calculated their individual errors (ΔPMI), and using those, we computed the average ΔPMI per measurement location, as depicted in Fig. 4f. With a mean ΔPMI of 0.26 h ± 1.38 h, the abdomen exhibits the lowest average reconstruction error as well as the smallest variation in error. The largest average ΔPMI and error variation were found for measurements conducted at the forehead, with a value of −1.46 h ± 2.88 h. The average of all errors, i.e., of all five measurement locations, was found to be 0.17 h ± 2.43 h. Moreover, 80% of all reconstructed PMIs, and 93% of the abdominal reconstructed PMIs, deviate no more than ±2.8 h (the smallest attainable uncertainty using the standard method) from the corresponding true PMI.

Next, we investigated whether a minimum difference between the body and the ambient temperature is required to achieve sufficiently accurate PMI reconstructions. The results of this

investigation are summarised in Fig. 5. This figure shows the average PMI reconstruction error as a function of the difference between body and ambient temperature (ΔT) per measurement location, where $\Delta T(PMI) = T_{body}(PMI) - T_{ambient}(PMI)$. For the abdomen, the chest and the upper arm, the average error remains within ±2 h for all ΔT, while for the forehead and the thigh larger errors are visible for ΔT below 5 °C and 1 °C, respectively.

**PMI reconstructions using visible and thermal 3D imaging.** Additionally, we reconstructed PMIs based on the scaled 3D models and co-registered thermal camera data of all bodies at the five body locations. Panels a–e of Fig. 4 show the PMIs reconstructed using these data as a function of their respective true PMIs per measurement location (camera-based data are shown as triangles). Again, to determine the accuracy of these reconstructed PMIs, we first computed their individual ΔPMIs, and using those, their average ΔPMIs per measurement location, as depicted in Fig. 4f. The lowest average PMI reconstruction error was found at the chest (0.58 h ± 1.47 h), while the largest average ΔPMI was found at the thigh, with a value of −1.05 h ± 1.73 h. The average of the errors of all five measurement locations was found to be −0.17 h ± 1.63 h. Moreover, 89% of all reconstructed PMIs, 90% of the reconstructed PMIs of the upper arm, and 83% of the abdominal reconstructed PMIs deviate no more than ±2.8 h from the corresponding true PMI.

**Field validation study.** Next, we performed RGB photogrammetry and recorded temperature logger data of bodies at four real crime scenes, including indoor and outdoor settings. We found that the necessary measurements for our approach were easily integrated within the existing forensic workflow at these varied crime scenes and that, on average, the total procedural time (i.e., measurements, TFD modelling and PMI reconstruction) did not exceed 15 min. Using the photogrammetry and temperature logger data, we reconstructed PMIs and compared them with PMIs determined by circumstantial information, such as telephone records and eye-witness statements. Figure 6 shows these reconstructed TFD-based PMIs (median of all body locations) as red circles for all four cases, while the red error bars denote the median absolute deviation. PMIs derived from circumstantial information are shown as blue lines. For all four cases, the predicted TFD-based PMI falls within the circumstantially determined timeframe for the PMI.

**Benchmarking against the current standard method.** Finally, rectal temperatures were recorded of the bodies at the four crime scenes. Using these data in conjunction with Henssge's nomogram (i.e., the standard method), the PMI was determined by an experienced forensic practitioner. Note that, in two of the four cases the body temperature at the time of death exceeded 37 °C (as assessed by the forensic medical examiner). Consequently, Henssge's nomogram is not applicable in these cases. For the first of the other two cases, the body was found at 08:35 and circumstantial information included an eye-witness statement according to which the deceased person was last seen alive at 03:00. The PMI predicted by the standard method was 11 h ± 2.8 h at the time of measurement (namely, 13:30) placing the estimated time of death between 23:42 and 05:18, while our method estimated the time of death as 04:20 with a possible range from 03:41 to 04:59. For the second case, the body was found at 10:15 and circumstantial information narrowed down the window for the time of death to 09:00–09:40. The PMI determined using the standard method was 5 h ± 2.8 h at the time of measurement (namely, 14:50), corresponding to a possible time of death between 07:02 and 12:38, while our method estimated the

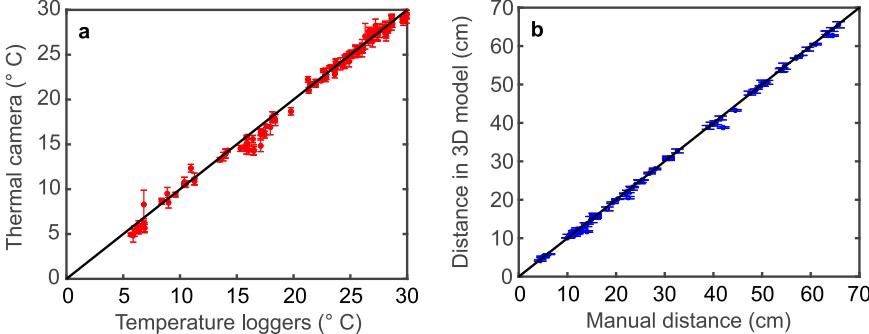

**Fig. 3 Benchmarking of the thermometers and the scaled 3D models. a** Comparison of the logger temperatures and their corresponding thermal imaging data (average ± standard deviation, n = 20). **b** Distances between markers on the human body measured manually and virtually in the scaled 3D models (error bars indicate the estimation errors of the virtual distance measurements in the 3D model). Black solid lines are a guide to the eye and denote the line of identity. Source data are provided as a Source Data file.

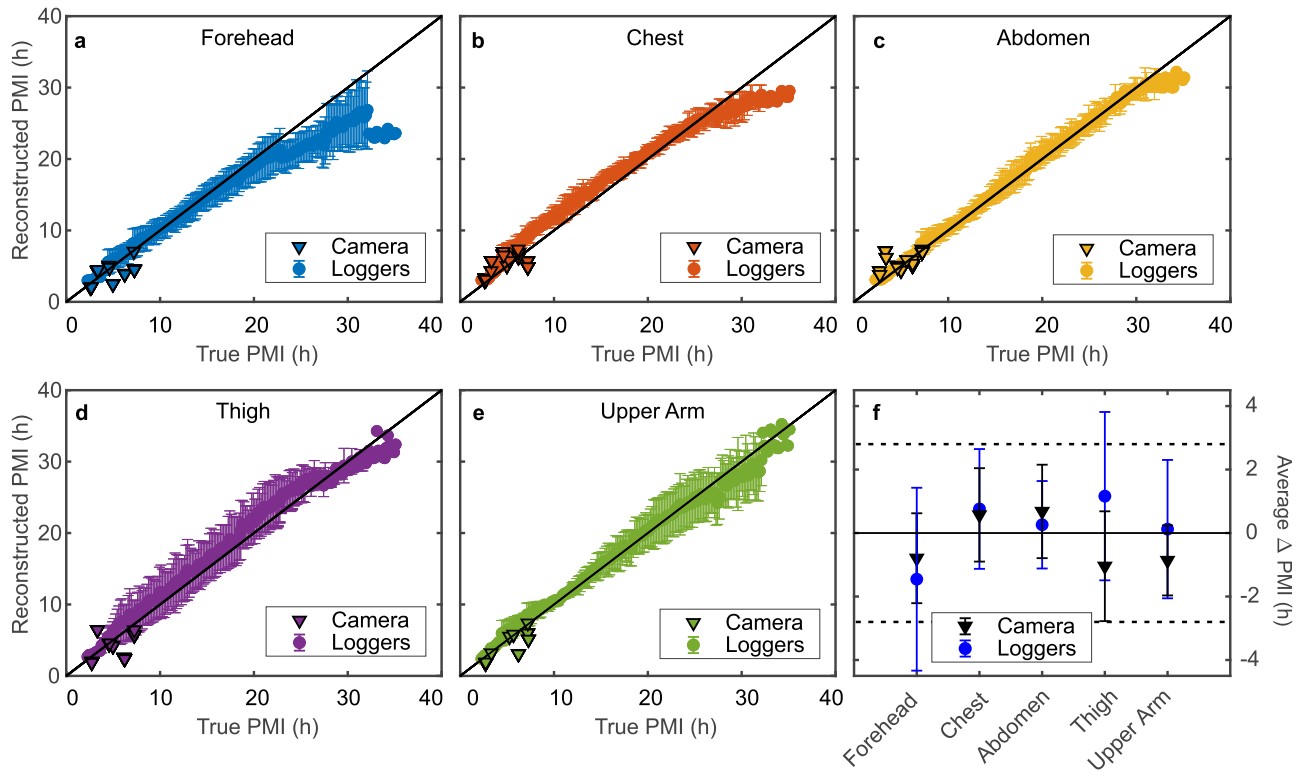

**Fig. 4 PMI reconstructions using our individualised TFD model.** PMIs reconstructed using the temperature logger (circles) and thermal camera (triangles) measurements from **a** the forehead, **b** the chest, **c** the abdomen, **d** the thigh, and **e** the upper arm versus their corresponding true PMIs (average ± standard deviation of six different bodies). The solid black line is a guide to the eye and denotes the line of identity. **f** Average error (ΔPMI) of all reconstructed PMIs shown in **a**–**e**, i.e., deviation from their corresponding true PMIs, per measurement location (error bars correspond to the standard deviation of data collected from six different bodies, n = 7167 for the logger data and n = 14 (forehead), n = 18 (chest), n = 18 (abdomen), n = 15 (thigh), n = 10 (upper arm) for the camera data). Solid and dashed black lines are guides to the eye, indicating errors of 0 h and ±2.8 h (smallest achievable error using Henssge's nomogram), respectively. Source data are provided as a Source Data file.

time of death as 09:09 with a possible range from 08:35 to 09:43. These data are also shown as PMI measured in hours in Fig. 6, where the PMI predictions by the standard method are shown as black lines (the relevant cases are indicated by an asterisk).

## Discussion

In this study, we developed an integrated computational approach for thermometric PMI reconstruction employing visible and thermal 3D imaging and validated it on recently deceased human bodies both in the morgue and at real crime scenes. We

first substantiated the feasibility of non-contact thermometry for our purpose, by comparing body temperatures measured using a thermal camera and temperature loggers, and found an average difference of −0.27 °C ± 0.61 °C between the two. The variation in these differences is most likely caused by changes in the imaging geometry, corresponding to variations in the actual physical size of the averaged regions of interest in the thermal images, and possibly local variations in surface temperature.

Next, geometric accuracy of the approach was determined by comparing manual and virtual reference measurements

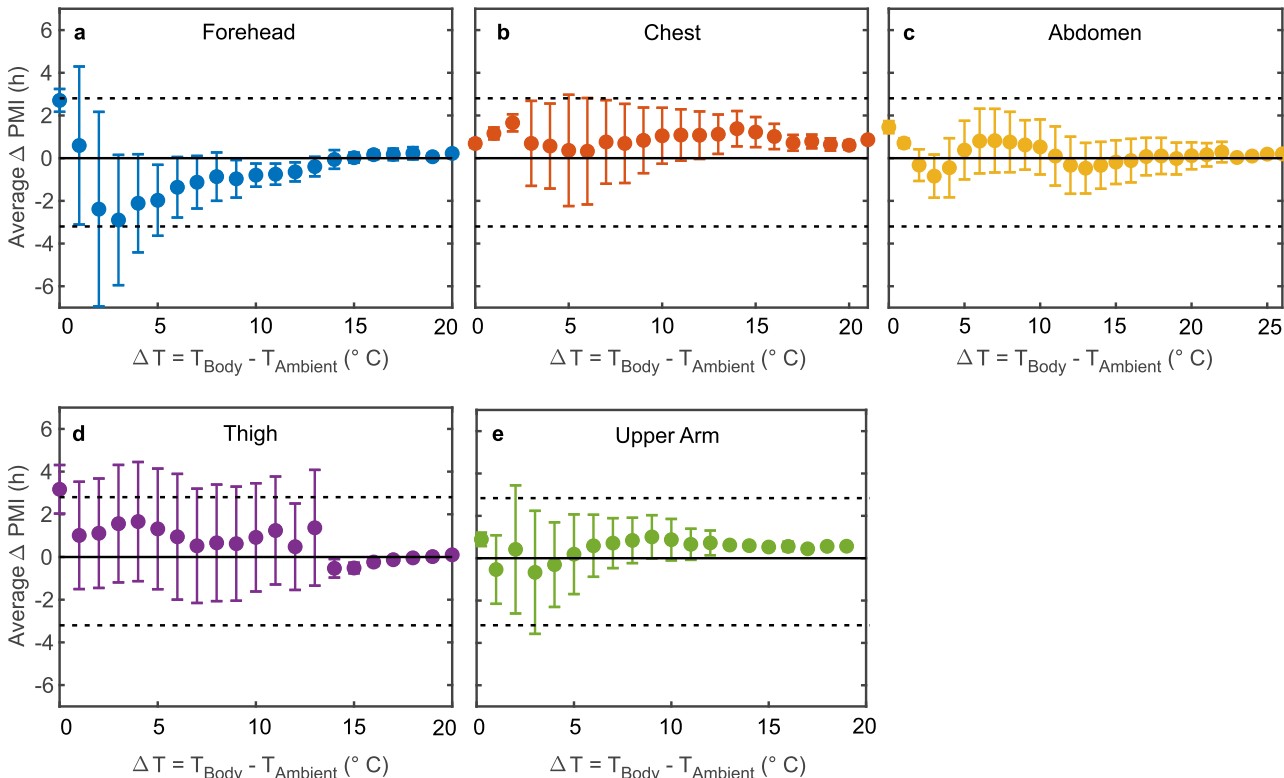

**Fig. 5 Influence of the difference between body and ambient temperature on the accuracy of the PMI reconstruction.** Influence of the difference between the measured body and ambient temperature (ΔT), i.e., the proximity to thermal equilibrium, on the error (ΔPMI) of the reconstructed PMIs (average ± standard deviation of six different bodies) calculated using temperature logger measurements from **a** the forehead, **b** the chest, **c** the abdomen, **d** the thigh, and **e** the upper arm. Solid and dashed black lines are guides to the eye, indicating errors of 0 h and ±2.8 h (the smallest achievable error using Henssge's nomogram), respectively. Note that at the thigh, the increase in standard deviation for ΔT below 13 °C is due to an increase in underlying data points, as these values of ΔT are more frequently encountered at this measurement location. Source data are provided as a Source Data file.

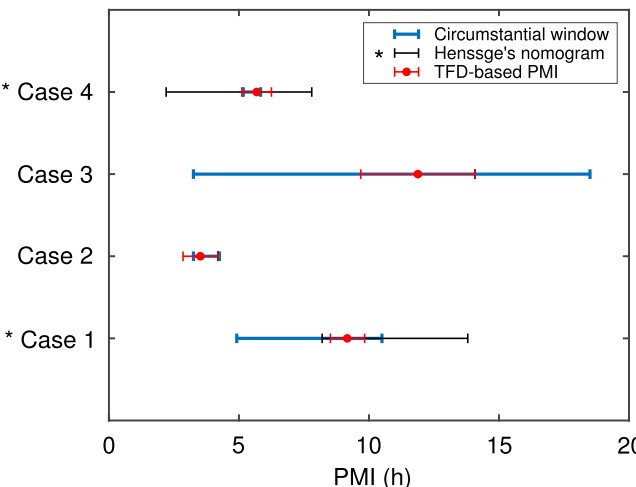

**Fig. 6 Field validation: PMIs reconstructed using our individualised TFD model in conjunction with measurements conducted at four crime scenes in both indoor and outdoor settings.** Red circles denote the median PMIs reconstructed using temperature logger data from all measurement (i.e., body) locations and visible photogrammetry. Red error bars correspond to the median absolute deviation. PMIs determined by circumstantial information (e.g., telephone records, eye-witness statements etc.) are shown as blue lines. For the two cases indicated with the asterisk (*), the black lines show the PMI determined using rectal and ambient temperatures measured at the crime scenes in conjunction with Henssge's nomogram (standard PMI-estimation method). Source data are provided as a Source Data file.

performed on the bodies as well as on scale bars placed in the scenes. We found the average difference between these reference measurements to be −0.3 cm ± 0.6 cm and 0.1 cm ± 0.2 cm, for the bodies and scale bars respectively. The cubic grid size of 1 cm$^3$ used for our TFD calculations, therefore, renders these scaled 3D reconstructions sufficiently accurate for our approach, as the geometric inaccuracies are smaller than the grid resolution. This, in turn, substantiates the capacity of our method to accurately measure the shape and posture of individual bodies.

The PMI reconstruction accuracy of our integrated method was assessed for two distinct measurement procedures. Here, we used our individualised TFD model in conjunction with skin temperatures measured at five body locations, either with (i) temperature loggers or (ii) a thermal camera, to reconstruct PMIs (of six subjects) ranging from 2 to 35 h. For PMIs reconstructed using data from the temperature loggers, we found the highest and lowest average error (ΔPMI) to be at the forehead and at the abdomen, with values of −1.46 h ± 2.88 h and 0.26 h ± 1.38 h, respectively. Moreover, 93% of the abdominal measurements yielded reconstructed PMIs within ±2.8 h of their true PMI. For PMIs reconstructed using data from the thermal camera, the largest and smallest average ΔPMI were encountered at the thigh and the chest, with values of −1.05 h ± 1.73 h and 0.58 h ± 1.47 h, respectively. Furthermore, of the reconstructed PMIs of the abdomen, 83% deviate no more than ±2.8 h from their true PMI. Together, these results (for both thermometer types) represent a notable improvement over the current standard method (Henssge's nomogram), where uncertainties range from ±2.8 h to ±7 h.

Additionally, using the temperature logger data, we investigated the extent to which these reconstruction errors depend on the difference between the measured body and ambient temperatures ($\Delta T$). These data may reveal a minimum temperature difference required to achieve sufficiently accurate PMI reconstructions. This, in turn, determines the range of applicability of our approach in forensic practice. Here, we benchmarked our approach against the highest achievable accuracy of the standard method, i.e., ±2.8 h for the reconstruction error. We found that for measurements carried out at the abdomen, chest and upper arm, average $\Delta$PMIs even within ±2 h are achievable for $\Delta T$ ranging from 1 °C to 26 °C. Notably, at the thigh, the variation in error increases for $\Delta T$ below 13 °C. This is caused by an increase in the number of underlying data points for these lower $\Delta T$, which are more frequently encountered at this measurement location. Accordingly, the thigh appears to be the measurement location least consistently yielding sufficiently accurate PMI reconstructions (compared with the standard method).

The largest deviations between reconstructed and true PMIs are encountered at the forehead, where they are mainly caused by underestimations of the true PMIs, i.e., an overestimation of the cooling rate at this measurement location. In our approach, we do not explicitly include the location of body cavities or bones, rendering the forehead one of the most simplified representations of the five measurement locations. Nevertheless, the overall results demonstrate that using our approach in conjunction with temperature measurements conducted at the abdomen or the upper arm achieves PMI reconstruction accuracies substantially outperforming the current standard method, even at extremely advanced stages of the heat exchange process, i.e., near thermal equilibrium.

Finally, the field validation study at four real indoor and outdoor crime scenes showed that, in all cases, the necessary measurements could be carried out and easily integrated within the existing crime scene workflow. Furthermore, PMI reconstruction using our approach (including measurement time) on average did not exceed ~15 min. For all investigated cases, the timeframe for the PMI was known through circumstantial information (e.g., telephone records, eye-witness statements etc.). We found that the TFD-based median PMIs all fell within the circumstantial timeframes. Moreover, we benchmarked our technique against the predictions of the standard method for two out of the four cases (as Henssge's nomogram was not applicable in the other two cases). Here, we found that our predictions fell within the timeframes predicted by the standard method and achieved a narrower timeframe for the time of death.

The above results demonstrate that SfM, coupled with advanced TFD modelling, can enable fast, accurate and non-invasive PMI determination for bodies of arbitrary shape and posture both in the lab and at real crime scenes. This, in turn, provides a practical and individualised approach for thermometric PMI reconstruction. Therefore, we believe that this is a promising approach, well placed to be used in forensic practice in the future. Nonetheless, we note that in this study, the practicality and accuracy of our approach was assessed in a limited number of settings. Consequently, determining the full range of applicability of this technique necessitates further validation studies testing the approach in other environmental conditions (such as climates, seasons etc.) and regions of the world. Similarly, given the limited number of samples in our benchmarking against the standard method, additional comparative validation studies (with sample numbers equalling those underlying the standard method) are required to fully establish the permissibility of our approach.

Despite the accuracy of the PMIs determined using our individualised TFD-model, we additionally note that the model is currently composed of only two distinct tissue types (adipose and non-adipose)[27]. Individualisation of the virtual body's tissue distribution could be achieved by including detailed tissue information derived from post-mortem computed tomography (CT) scans[36,37]. This tomographic information would also enable the modelling of body cavities filled with air, fluids or other materials, addressing issues such as that of the forehead mentioned above. Integration of such tomographic data may improve model accuracy; however, by the same token, it may also curtail practical applicability of the approach: the increased number of model minutiae would manifest as a higher computational workload, larger data sets and possibly decreased ease-of-use. Notwithstanding, the prevalence of forensic post-mortem CT scans is steadily increasing and the required information may therefore be available in a larger proportion of forensic cases in the future.

Further improvement in the accuracy of our PMI reconstructions may be achieved by a reduction in the uncertainties of the model input parameters. To this end, thermal properties, e.g., thermal conductivity of the materials in contact with the body, and hence relevant to the heat exchange problem, could be determined directly at the crime scene. Another model parameter likely to vary in forensic practice, and thereby increase the uncertainty in the time of death estimate[38], is the ambient temperature. In many cases, however, this information may be available as thermostat or meteorological data, allowing easy integration in the thermodynamic computations. Moreover, a general strength of our individualised TFD model is its capacity to simulate various scenarios (e.g., temporally varying ambient conditions such as changing surrounding media and temperatures) to reconstruct a whole range of possible PMIs. The minimum and maximum of this range can then be reported as the most likely timeframe for the PMI. Furthermore, our method is compatible with parameter optimisation strategies. Here, unknown model parameters (such as ambient temperature, air flow etc.) are varied until the simulated temperature evolution best describes the measured one (i.e., until the difference between the simulated and the measured temperature series is minimal).

A conceivable practical limitation is posed by crime scenes involving movement restricting surroundings. Such circumstances may hamper the necessary photogrammetric measurements and hence the generation of sufficiently accurate 3D models (or prevent their creation altogether). In our field validation study, however, we were able to generate sufficiently detailed 3D models even in movement limiting circumstances, such as a narrow bedroom. Moreover, we note that in all investigated cases *rigor mortis* preserved the body posture. In such cases, photogrammetric measurements could therefore also be carried out once the body has been moved to more spacious surroundings.

Comparing both measurement approaches for our technique, the thermal camera-based implementation provides a much larger number of separate temperature measurements. These, in turn, could be used to derive more densely sampled PMI distributions. As a result, the outcomes of the thermal camera-based implementation (if combined appropriately) could yield even more robust PMI determinations. In addition, as thermal images retain spatial information, they contain more case-specific information, which may be useful at a later stage of the investigation. At the same time, the currently much higher procurement costs for the necessary hardware may hamper widespread adoption of the thermal camera-based approach in the forensic field. Likewise, utilising the unreduced thermal photogrammetry data set will result in increased computation times, potentially limiting acceptance by crime scene investigators. In contrast, the use of the less costly thermal loggers is simple and requires no additional training, in turn, likely facilitating rapid adoption by forensic

practitioners. Technological strides may, however, improve the speed and affordability of the full-body thermal photogrammetry approach in the future.

Taken together, the findings of this study show that our approach is uniquely optimised for potential future application in forensic practice in three distinct ways. First, by measuring the exact shape and posture of the body, it enables TFD-based PMI reconstruction for bodies in arbitrary postures. This, in turn, may allow application of this approach in a much wider range of forensic cases, where bodies are often found in non-straight postures. Second, the use of thermal photogrammetry provides, for the first time, a non-contact means for thermometric PMI determination. This has important forensic implications, as non-contact methods prevent contamination and destruction of other traces. Third, it reduces the required measurement time from ca. 45 min to ca. 15 min, which is expected to facilitate its integration within existing crime scene investigation protocols. In conclusion, our approach enables highly location-specific comparison of measured and simulated post-mortem temperatures of bodies in arbitrary postures. This, in turn, allows accurate thermometric PMI reconstruction with minimal user input and the lowest errors for PMI reconstruction to date. Together, these results suggest a potentially broad range of future applicability and hence forensic relevance of individualised TFD-based PMI reconstruction.

## Methods

**Bodies**. Human cadavers were obtained through the body donation program (BDP) of the Department of Medical Biology, Section Clinical Anatomy and Embryology, of the Amsterdam University Medical Centers (UMC), location Academic Medical Center (AMC), in the Netherlands. All experiments were performed in accordance with international and institutional ethics guidelines, in accordance with Dutch legislation and the regulations of the medical ethical committee of the Amsterdam UMC at the location Academic Medical Center. The Dutch Burial and Cremation Act ('Wet op de Lijkbezorging', WLB; Article 1 and Article 67) describes donation to science as one of the three possible final destinations of human remains (the other two being burial and cremation). The procedure of body donation to the Department of Medical Biology of Amsterdam University Medical Centers (Amsterdam UMC)- location AMC and the subsequent use of these bodies for scientific research is not subject to medical ethics review. The Dutch Act on Human Subject Research does not cover research with bodies, donated in the context of the Burial and Cremation Act. Moreover, Dutch legislation does not have other regulations in place that require review in case of concrete research protocols with donated bodies. The procedure for body donation was approved by the Department of Health Law of the AMC. The Medical Ethics Committee of the AMC, provided a waiver for individual studies that make use of donated bodies. The protocols, the methodology, the academic merits, eventual privacy and ethical issues, and environmental issues were reviewed by the Department of Medical Biology, being responsible for the BDP. Written consent (as an extension of an existing codicil) was obtained from all donors, in which they agreed to the use of their bodies and information in taphonomic studies and the publication thereof. In addition, consent to publish (in compliance with the tenets of the Declaration of Helsinki) was collected from the living human subject shown in Figs. 1a, c, d and 2. The time of death was determined by a physician. One of the deceased persons underwent euthanasia. In all other cases the cause of death was old age/natural causes. Following death, the bodies were immediately transported to the hospital morgue as per protocol of the BDP[39]. In the morgue validation study, data were collected from six bodies (three male and three female). The subjects' ages, weights and heights ranged from 68 years to 87 years, from 44 to 71 kg, and from 152 to 177 cm, respectively. Upon completion of the SfM measurements, all bodies were stored in a 2–6 °C refrigerated chamber, except for one body, which was kept at room temperature for the entire duration of the temperature logger measurements. Available measurement time points, i.e., PMIs, were dictated by the time of arrival of the body at the morgue and ranged from 2 to 35 h post-mortem. The ambient temperatures at the morgue, as well as in the refrigerated chamber used to store the body, were recorded for the full duration of the body temperature measurements.

**Preparation and experimental set-up**. First, any coverage, such as cotton sheets, around the body and clothing was removed in order to expose five thermometry locations (the forehead, the chest, the abdomen, the thigh and the upper arm). We then recorded subject-related data including age, weight, time of death and body dimensions. Next, reference scale bars comprising coded imaging targets were placed beside the body, as shown in Fig. 1b. This step provided a coordinate reference frame, essential for accurate reconstruction of the body geometry in

physical units [m]. Following this, the temperature loggers, shown in Fig. 1a, (Thermochron iButtons, Maxim Integrated, San Jose, CA, U.S.) were activated (OneWire Viewer 1.0 Software, Maxim Integrated, San Jose, CA, U.S.) and placed on the forehead, the chest, the abdomen, the thigh and the upper arm. One additional temperature logger was placed next to the body to measure the ambient temperature. All sensors were marked with a unique coded imaging target (marker) (Fig. 1b), allowing automatic registration of their respective positions within the scaled 3D model of the body (Fig. 1d). Temperature logging was performed every 60 s and over several hours. Total measurement durations ranged from 16 to 24 h.

**Photogrammetry**. Two distinct approaches were evaluated. In the first (Approach 1), photographs were recorded using a Nikon D5300 Digital Single Lens Reflex camera. In the second approach (Approach 2), a FLIR T540 thermal camera (FLIR Systems, Inc., Wilsonville, USA) operating in the wavelength range from 7500 nm to 14500 nm, with a 42° lens was utilised. This camera simultaneously records $464 \times 348$ pixel thermal images and $1280 \times 960$ pixel visible RGB images. Here, we used the FLIR Tools+ 5.13 software to generate RGB images of identical resolution and field-of-view as the thermal images. Moreover, using the same software, thermal images were converted to temperature maps, containing the measured temperature values instead of false colour pixel values.

**Generating and exporting the scaled 3D model**. Using the series of overlapping RGB photographs recorded in the previous step, scaled 3D models of all bodies were generated using Agisoft PhotoScan Professional 1.2 software (Agisoft LLC, St. Petersburg, Russia). This software aligns the series of recorded RGB images and, using the aligned images, computes a point cloud comprising the three-dimensional coordinates for all reconstructed surface points. These coordinates are expressed in physical units [m] using the reference frame of the coded scale bar. Here, we used the standard settings suggested by the software for every processing step in the work flow. In this way, we generated point clouds with average densities (±standard deviation) of $32 \pm 25$ pts/cm$^3$ and $31 \pm 57$ pts/cm$^3$, for the morgue data and the crime scene data respectively. Next, we generated and exported triangle surface meshes of the scaled 3D models enabling individualisation of the TFD model (Fig. 1d). By generating a surface mesh as the topological boundary of the point cloud, we gained access to its topological properties, such as the outer-pointing normal. This, in turn, allows us to accurately model postures resulting in non-convex hulls of the point cloud (i.e., body postures which result only in partial contact with the substrate, such as bent limbs, arched small of the back etc.) by providing a definition of "inside/outside" of the point cloud. Moreover, this representation reduces spatial noise and file size and facilitates memory management.

**Individualising the TFD model**. Volumetric representations of the bodies within TFD 3D cubic grids were created by means of a custom-written code (MATLAB 2018b, The Mathworks Inc., Natick, MA, USA) which sequentially determines the corresponding grid location for all points in a bodies' surface mesh (Fig. 1d). As this surface mesh inherently contains the interface between the body and the substrate/ground, assignment of material types, i.e., human tissue or substrate/ground, and their corresponding thermal properties is straightforward.

**Co-registration of the body geometry and temperature**. The specific locations of the temperature loggers on the body in Approach 1 were labelled by the coded imaging targets, which were detected automatically. To extract the temperature measurements for Approach 2, all thermal images depicting these detected targets were selected.

**TFD model**. PMI reconstruction using our approach necessitates spatially-resolved simulation of the change in skin temperature as a function of the PMI. To this end, 3D heat exchange simulations were carried out for all bodies using a TFD model (Fig. 1e)[27]. Thermal properties (e.g., thermal conductivity, specific heat capacity etc.) are assigned to all cubes in the grid (depending on which material they represent, e.g., adipose or non-adipose human tissue, clothing, air, substrate etc.) as well as an initial temperature (e.g., 37 °C for the body and 20 °C for the environment). Next, a temporal step size is defined (in our case 60 s) and by applying the laws of thermodynamics describing conductive, convective and radiative heat transfer[27], the amount of heat energy exchanged between neighbouring cubes during one temporal step is computed. This, in turn, yields the corresponding change in temperature for every cube, which is then used to update their current temperatures. By repeating this calculation for consecutive time steps, we are able to simulate the temperature of every cube in the grid for every point in time, i.e., spatially-resolved body temperatures as a function of the PMI (Fig. 1e). Note that, using this approach, we can easily account for changes in the ambient temperature by simply adjusting the current temperatures of the relevant cubes. Indeed, in this study, we used the measured ambient temperatures to set the corresponding temperatures in the simulations.

**PMI reconstruction using temperature loggers**. All measured temperature series (of all five locations on all six bodies, i.e., 30 in total) comprised between 960 and 1440 datapoints (corresponding to total measurement durations between 16 and 24 h). We reconstructed PMIs for all of these individual data points.

**PMI reconstruction using a thermal camera**. We computed temperatures from the thermal images of each of the five measurement locations of all bodies. Here, the number of data points (i.e., temperatures) per body location for each of the six bodies ranged from one to three, yielding a combined total of 10–18 measurements for each body location. The true PMIs of these measurements ranged from 2 to 7 h. Analogous to the data processing of the temperature loggers, we reconstructed PMIs for all individual data points.

**Forensic field validation in indoor and outdoor settings**. We tested our approach in four different forensic cases including indoor and outdoor settings. All measurements were performed during official crime scene investigations carried out by the police and therefore, according to Dutch law, fall under the jurisdiction of the public prosecutor who gave permission to collect and publish the reported data. The circumstances of death included hanging in a wardrobe, fever following hypothermia treatment in a hospital bed, cardiac arrest following cardiopulmonary resuscitation on a narrow bedroom floor and severe impact traumata following a jump from the 6th floor of a building. The deceased included two males and two females with weights and heights ranging from 50 to 115 kg, and from 160 to 174 cm, respectively. Following the completion of the standard crime scene protocol, the forensic practitioner activated and attached five thermal loggers to the body. An additional logger was placed in the vicinity of the body to record the ambient temperature. All thermal loggers were sterilised/decontaminated before each use. In cases where the body was clothed, small incisions were made in the fabric (without moving or undressing the body) to ensure contact of the thermal loggers with the skin (or exposure of the skin for non-contact thermometry using a thermal camera). The loggers were then labelled with coded imaging targets and photogrammetric measurements (total measurement time ~5 min) were carried out using a Nikon D850 Digital Single Lens Reflex camera. In the case of hanging in the wardrobe, the body was moved to enable the photogrammetric measurements. We found that the posture of the body was preserved by *rigor mortis*. 3D models were then generated from these series of RGB images (according to the protocol described above) and, using the 3D models, individualised TFD simulations were carried out for all four bodies. Generation of the 3D models took ~5–7 min, while the TFD simulations were completed with runtimes of ~20 s. The length of the measured temperature series varied between cases and ranged from 6 to 53 datapoints (6–53 min). We reconstructed PMIs for all of these individual measured temperatures by comparing them to their co-registered TFD simulations. As the last step, we grouped the reconstructed PMIs from all measurement (i.e., body) locations per case and from that determined the median PMI as well as the median absolute deviation. These then yielded the case-specific estimated PMI and possible range for the PMI, respectively.

**Benchmarking against the standard method**. In all four forensic cases, rectal temperatures and body weights were determined. In two cases, the body temperature at the time of death exceeded 37 °C (as determined by the forensic medical examiner) and Henssge's nomogram was therefore not applicable. For the other two cases, these data were used in conjunction with Henssge's nomogram (the current standard PMI-estimation method) and PMIs were determined by an experienced forensic practitioner. These PMIs, in turn, served as reference PMIs in the benchmarking of our approach against the current standard PMI-estimation method.

**Reporting summary**. Further information on research design is available in the Nature Research Reporting Summary linked to this article.

## Data availability

The authors declare that all data supporting the findings of this study are present within the paper. The raw data are protected and not available due to data privacy laws. The source data underlying Figs. 3a-b, 4a–f, 5a–e, and 6 are provided as a Source Data file. A minimum dataset for the interpretation, verification and extension of the research can be accessed at https://doi.org/10.5281/zenodo.5070592[40]. Source data are provided with this paper.

## Code availability

The MATLAB scripts can be accessed at https://doi.org/10.5281/zenodo.5070592[40] and used for academic purposes only.

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

## Acknowledgements
The authors would like to thank N. Islam for her help in carrying out measurements, A. van Dam for her design of a figure, L. Aalders for his 3D printing of the imaging targets, the crime scene investigators of the Forensic Investigations Department of The Hague police force (Forensische Opsporing Politie Den Haag) for granting us access to the crime scenes and the participants of the body donation program for their contribution to the advancement of science. Funding sources: Project 'Therminus' is funded by the Innovation team of the Dutch Ministry of Justice and Security.

## Author contributions
L.S.W. and G.J.E. developed software, performed measurements and analysed data. M.R. performed the field validation measurements. M.C., I.D. and J.V.M. performed/assisted measurements. R.O. provided material. M.C.G.A. performed measurements, conceived and supervised the project. All authors contributed to planning, design of experiments, discussion of results, and writing of the manuscript.

## Competing interests
The authors declare no competing interests.
