## [Peer Review File · Nature Communications]

Individualised and non-contact post-mortem interval determination of human bodies using visible and thermal 3D imagingReviewers' comments:

Reviewer #1 (Remarks to the Author):

The authors propose a method that uses a combination of photogrammetry, thermal imaging and a numerical analysis so simulate post-mortem heat dissipation and estimate the post-mortem interval. While I do have expertise in forensic and post-mortem imaging, I am not an expert when it comes the post-mortem interval as I have a technical background, not a medical one. I do find the article well written and to me the proposed method, from a technical standpoint, makes sense. There are however a couple of comments I have with regards to the bodies used in this study and the practicality of the approach:

- 1: How was the actual time of death determined and by whom? What is the potential error here? Was someone in the room with the dying person?
- 2: What was the cause of death and how was it determined? Was the body temperature measured right after death?
- 3: What was the body temperature at the time of death? Was there fever or hypothermia or were drugs or medications involved that potentially alter the body temperature?
- 4: Were the bodies moved post-mortem, i.e. for undressing or for transport or were they scanned at the place of death? If they were moved, this would go against the premise of developing a technique that allows simulating PMI for bodies found in non-straight positions because it skews the results. This could be mentioned as limitation.
- 5: Was the room temperature at the place of death / during transport / in the morgue measured? Was the time measured, the bodies were exposed to these temperatures?
- 6: The authors should discuss limitations of the method in more detail. For instance: while it might be, that the proposed method is more accurate than the Henssge formula, it is also more impractical and slower. At our Institute, every forensic pathologist has an app on their phone, which allows them to calculate the PMI within seconds at the scene. This means, the state attorney and police can get information about the possible time of death at once. Often, bodies are otherwise covered, i.e. by sheets or garbage (i.e. in case of hoarders).
- 6: In the discussion, the authors write that the results demonstrate that their method enables an "accurate PMI determination for bodies of arbitrary shape and posture...". This might be a bit problematic in a real forensic setting, because most deceased will have to be undressed, which changes the position and thus the simulation. It can also happen, that the body cannot be properly scanned, if the position it is found in is too narrow.
- 7: For assessing practicality, it would also be informative to know how long the entire pipeline takes. How long does imaging take, how long processing of the images and how long the simulation (about 20 seconds according to the previous publication)?
- 8: One point where this method might be very valuable is the PMI for children and infants, as these are not covered by the Henssge formula as far as I know.
- 9: The authors suggest CT scanning to improve the simulation. For future work, they might want to look at MRI scanning, as it is possible to estimate the temperature from the image contrast. This way, the entire estimation could be fully automated.
- 10: The exact settings for camera and the photogrammetric reconstructions would increase the reproducibility.

While I do not agree with the authors when it comes to practicality and feel, that a bit more information about the cases is required, to me the approach is interesting and innovative and could be interesting for the journal after some revisions.

Lars Ebert

Reviewer #2 (Remarks to the Author):

The major claim of the paper is the introduction of a non-contactless measurement technique in the forensic field that determines the shape and skin temperature distribution of bodies and utilizes a subsequent simulation using a thermodynamic finite-difference (TFD) algorithm for estimating the post-mortem interval (PMI).

The novelty of the contribution lies less in the methodology of the used processing steps like 3d reconstruction by structure from motion, thermal mapping and co-registration, or simulation of heat transfer, but more in the proposed combination of the involved techniques for improving the estimate of the PMI and the solid investigation on the applicability in the forensic domain. The study presented here based on photogrammetry, thermography, and TFD modelling is convincing and gives a new view to the addressed application field.

The approach is also interesting in a wider field, because it follows the general trend to exploit technologies from photogrammetry for capturing individual shapes of objects by point clouds and using these descriptions seamless for various numerical simulations based on finite cells, see e.g. an application in a different field, namely structural analysis (Kudela et al., 2020) [doi: 10.1016/j.cma.2019.112581].

Minor reviewer comments (R) to authors statements (A):

A1: (074) The authors state "... our TFD model accurately simulates the heat exchange between the body and its environment, yielding spatially-resolved simulations of the body temperature as a function of the PMI."

R1-1: Generally, heat transfer happens by conduction, convection, and radiation. Please, name the components that are modelled in your "accurate" simulation.

R1-2: In case your model considers conduction, please clarify to which extend the ground is modelled (area, material, etc.)!

R1-2: In case your model considers convection, please name the parameters that are taken into account!

R1-3: In case your model considers the radiation transfer, please clarify to which extend is the surrounding environment modelled?

A2: (076) "The time point at which the simulated temperature best approximates the measured temperature (at the same body location) then corresponds to the reconstructed PMI."

(119) "Finding the time point at which the simulated skin temperature best approximates its spatially co-registered measured skin temperature then corresponds to the reconstructed PMI ..."

R2: Please, give a short remark how do you determine (calculate) the best approximation of two spatial temperature distributions (measured/simulated)!

A3: (108) "Next, we individualised our TFD model by first generating and then translating triangle surface meshes of the scaled 3D models into volumetric representations of the bodies within 3D cubic grids."

R3: Please, explain why you first generating a triangle surface mesh. The point cloud from SfM can directly transferred into a volumetric representation (voxels).

A4: (284) First, any coverage, such as cotton sheets, around the body and clothing was removed in order to expose five thermometry locations ... "

(056) "Second, as this model uses core temperatures to reconstruct PMIs, it intrinsically relies on rectal thermometry, increasing the risk of trace contamination and destruction."

(263) "This has important forensic implications, as non-contact methods prevent contamination and destruction of other traces. Third, it reduces the required measurement time from ca. 45min to ca. 10min, facilitating its integration within existing crime scene investigation protocols."

R4-1: Please, clarify for the practical (operational) use of the proposed method: You are considering skin temperatures, right? Has the entire clothing to be removed before thermography?

R4-2: If YES, how do you prevent contamination and destruction of other traces?

R4-3: If YES, I this the required time included in the 10min?

R4-4: If NO, how do you consider the clothing in measurement and simulation?

A5: (335) "Note that, using this approach, we can easily account for changes in the ambient temperature by simply adjusting the current temperatures of the relevant cubes."

R5-1: Please, address briefly the challenges and limitations of the approach at the end of the paper and distinguish between indoor and outdoor simulations.

R5-2: Please, address for outdoor simulation the problem of parameter setting for the post-mortem interval with unknown starting situation (time of death). During the PMI the parameter set

of the outdoor environment typically changes over time, e.g. in the morning: increasing radiation by sun, ground temperature, temperature of nearby objects, wind, humidity, rain, etc.)

A6: (094) "We first established two orthogonal approaches to obtain co-registered anthropometric and thermometric data of deceased human bodies, which then served as modelling inputs for TFD-based PMI reconstructions."

R6: Could you give a statement based on the comparison of both approaches concerning their potential for further investigations? Is it an advantage that thermography provides the temperature distribution over the entire hull of the body instead of just measuring single points by temperature loggers?

A7: (193) "We found the average difference between these reference measurements to be $-0.3\text{cm} \pm 0.6\text{cm}$ and $0.1\text{cm} \pm 0.2\text{cm}$, for the bodies and scale bars respectively. The cubic grid size of 1cm^3 used for our TFD calculations therefore renders these scaled 3D reconstructions sufficiently accurate for our approach, as the geometric inaccuracies are smaller than the grid resolution.

(301) "This camera simultaneously records 464×348 pixel thermal images and 1280×960 pixel visible RGB images. Here, we used the FLIR Tools+ software to generate RGB images of identical resolution and field-of-view as the thermal images.

R7: Please, could name the point density (pts/cm³) of the point cloud which was received from the adapted RGB (464×348 pixel) of the FLIR camera?

Reviewer #3 (Remarks to the Author):

The authors have tapped into a unique set of procedures for estimating the PMI of human remains. As pointed out in their manuscript, determining the PMI accurately and precisely remains a massive challenge for the forensic sciences community. I found the manuscript to be well-written, concise, and thought provoking. However, I also recognized several limitations to the study that the authors should address. The two major limitations of the current study are, 1) lack of sample size from their location or from others around the world, and 2) no validation.

Response to reviewers' comments

Below we respond in detail to each of the reviewers' comments and describe how we have adapted our manuscript accordingly. For the benefit of all reviewers, before addressing the specific concerns, we first provide a general response where we highlight a new field validation study that we have performed. All changes, including the results of the field validation, are highlighted in red text in the revised manuscript.

General response to all reviewers

Although we validated our method on human bodies, these studies were performed in a morgue environment. We therefore accept that the applicability and practicality of our method in the context of real forensic fieldwork was not explicitly demonstrated. To address this concern, we have undertaken a collaboration with the Dutch National Police and the Netherlands Forensic Institute. In this study, we applied our method to estimate the time since death of human bodies found at various indoor and outdoor crime scenes, *i.e.*, where the temperatures and 3D shapes of the bodies were measured *in situ*. Crucially, for all four cases investigated, the actual time since death was known through circumstantial information. Causes of death ranged from hanging, extended hyperthermia, cardiac arrest following cardiopulmonary resuscitation and severe impact-related traumata. As a control, we also recorded rectal temperatures at the scene to use in a benchmarking step against the gold standard. This unique field study has allowed us to fully validate our method in real crime scene settings. In this way, we have demonstrated that (i) our method is easily integrated in the standard crime scene investigation workflow and (ii) our approach for post-mortem interval determination has similarly high accuracy for bodies found at real crime scenes as those in our morgue validation experiments. Importantly, these accuracies are significantly higher than the gold standard. We believe that our new validation at real crime scenes represents a major advancement for our approach and provides even greater confidence in its wider applicability and practicality.

Response to Reviewer #1

Comment 1

How was the actual time of death determined and by whom? What is the potential error here? Was someone in the room with the dying person?

The time of death was determined by a physician. One of the deceased persons underwent euthanasia and the time of death was therefore known precisely. We have amended the manuscript to state this more clearly (see lines 352-353).

Comment 2

What was the cause of death and how was it determined? Was the body temperature measured right after death?

In all morgue validation cases, the cause of death was old age/natural causes and in one case euthanasia. For these cases, the body temperature was not measured right after death as this is not part of the standard medical protocol for death from natural causes. We have revised the manuscript to state this more clearly (see lines 353-355).

Comment 3

What was the body temperature at the time of death? Was there fever or hypothermia or were drugs or medications involved that potentially alter the body temperature?

In the morgue validation study, there was no evidence to suggest that any of the deceased persons exhibited peri-mortem hypo- or hyperthermia. In two cases of the field-validation group, however, a higher peri-mortem temperature was assumed: in one of these cases, the victim was hospitalised at the time of death and therefore the peri-mortem body temperature was monitored (showing hyperthermia) and in the other case, peri-mortem hyperthermia was deduced on the basis of the rectal temperature by the medical forensic examiner. This information has been added to the Methods section of the revised manuscript.

Comment 4

Were the bodies moved post-mortem, i.e., for undressing or for transport or were they scanned at the place of death? If they were moved, this would go against the premise of developing a technique that allows simulating PMI for bodies found in non-straight positions because it skews the results. This could be mentioned as limitation.

For the morgue-based validation study, bodies were transported to the hospital morgue in accordance with our hospital's body donation programme protocol. While the morgue indeed represents a controlled experimental environment, some of the bodies arrived at the morgue in non-straight postures (see for instance Figure 1 and Figure 2 of the manuscript), which allowed us to measure on different body shapes even in this setting. However, we agree with the reviewer that in order to establish the method's applicability in forensic practice, our approach should be evaluated under more realistic circumstances. To address this, we have now carried out a field validation study (see section titled 'General response to all reviewers' above). The results of this new study demonstrate that our method also provides accurate PMI reconstructions under such uncontrolled conditions/realistic circumstances. We have included the outcomes of this new field validation in the revised manuscript.

Comment 5

Was the room temperature at the place of death / during transport / in the morgue measured? Was the time measured, the bodies were exposed to these temperatures?

While the ambient temperatures at the place of death and during transport were not measured, the ambient temperature at the morgue as well as in the refrigerated chamber used to store the body were recorded for the full duration of the body temperature measurements. We have added this information to the manuscript (see lines 361-363). As bodies included within the body donation programme usually die in indoor settings (at home), we have no reason to believe that the ambient temperatures deviated significantly from common room temperatures. Moreover, the geographical inclusion radius of the body donation programme of our hospital is small. As a result, transport durations for the bodies (and hence exposure to varying ambient temperatures) are short. Indeed, bodies arrived at the morgue on average 5h after the nominal time of death (some even as soon as 1.5h post-mortem). The fact that, despite these unknown variations, our method yields accurate reconstructions of several individual PMIs suggests that the influence of these variations was sufficiently small.

Comment 6

The authors should discuss limitations of the method in more detail. For instance: while it might be, that the proposed method is more accurate than the Henssge formula, it is also more impractical and

slower. At our Institute, every forensic pathologist has an app on their phone, which allows them to calculate the PMI within seconds at the scene. This means, the state attorney and police can get information about the possible time of death at once. Often, bodies are otherwise covered, i.e., by sheets or garbage (i.e., in case of hoarders).

Our new field validation study at real crime scenes (see also the section titled ‘*General response to all reviewers*’ above) revealed that a total procedural time of ~15min (incl. photogrammetry and thermometry) was sufficient for accurate PMI reconstructions. We agree with the reviewer that the use of an app is fast and convenient in many cases. However, all of these apps utilise Henssge’s nomogram to estimate the PMI and are therefore subject to the same limitations: (i) large and non-deterministic error margin; (ii) invasive rectal thermometry (risk of trace contamination); (iii) operator-dependent, i.e., subjective, use (especially in cases involving coverage and hence correction factors). All of these limitations are easily and uniquely overcome with our method. With regard to coverage, any type of thermally participating materials (e.g., coverage) are easily included in the TFD model. Moreover, similar to our method, the above-mentioned apps require thermometric data, the recording of which adds to the total duration of the procedure (i.e., longer than mere seconds). We therefore believe that our method is both more accurate and highly practical.

Comment 7

In the discussion, the authors write that the results demonstrate that their method enables an “accurate PMI determination for bodies of arbitrary shape and posture...”. This might be a bit problematic in a real forensic setting, because most deceased will have to be undressed, which changes the position and thus the simulation. It can also happen, that the body cannot be properly scanned, if the position it is found in is too narrow.

In order to assess the practicality of our approach, especially with regard to the concerns raised here by the reviewer, we have applied our method at a variety of real (indoor and outdoor) crime scenes (see also the section titled ‘*General response to all reviewers*’ above). These crime scenes included movement limiting environments (e.g., narrow bedroom, shipping container) for which we found that 3D imaging was still possible. Moreover, we found that the body posture was conserved by *rigor mortis* even after movement of the body. 3D scans could therefore also be conducted after moving the body to a less movement-limiting space. Furthermore, most of the evaluated cases involved clothed bodies. In order to enable the necessary thermometry, small incisions were made in the clothing (without undressing or moving the body) through which the thermal loggers were attached to the skin. We have added this practical information to the Method section of the revised manuscript.

Comment 8

For assessing practicality, it would also be informative to know how long the entire pipeline takes. How long does imaging take, how long processing of the images and how long the simulation (about 20 seconds according to the previous publication)?

We agree with the reviewer that this information would be beneficial to the reader and have therefore added the following information to the Method section of the manuscript. Results of our field validation study showed that the duration of the entire pipeline remained around ~15min. The acquisition of the photogrammetric data takes as little as ~5min. The total simulation time will vary with body dimensions and number of time-steps. For the field validation study, we found total runtimes of ~20s for simulations of the cooling of average-sized bodies over a 24h period (i.e., 1440 time-steps). Currently, the most time-consuming step is the generation of the 3D model which on average took ~5-7min. This step can in principle be expedited by means of other 3D surface technologies.

Comment 9

One point where this method might be very valuable is the PMI for children and infants, as these are not covered by the Henssge formula as far as I know.

We agree with the reviewer that our approach may be of particular interest in this domain and are grateful for this constructive feedback.

Comment 10

The authors suggest CT scanning to improve the simulation. For future work, they might want to look at MRI scanning, as it is possible to estimate the temperature from the image contrast. This way, the entire estimation could be fully automated.

We thank the reviewer for this interesting and valuable suggestion.

Comment 11

The exact settings for camera and the photogrammetric reconstructions would increase the reproducibility.

We agree that this information aids reproducibility and have therefore amended the manuscript accordingly (see lines 392-393).

Comment 12

While I do not agree with the authors when it comes to practicality and feel, that a bit more information about the cases is required, to me the approach is interesting and innovative and could be interesting for the journal after some revisions.

We thank the reviewer for this constructive feedback. We believe that our additional field validation study, along with our other improvements to the manuscript (outlined in detail above), address the reviewer's concerns regarding both practicality and validation.

Response to Reviewer #2

Comment 1-1

Generally, heat transfer happens by conduction, convection, and radiation. Please, name the components that are modelled in your "accurate" simulation.

Our TFD model incorporates all three heat exchange pathways. This is detailed in our previous paper (Sci. Adv. 6, eaba4243 (2020), doi:10.1126/sciadv.aba4243). We have revised the current manuscript to state this more clearly (see line 421).

Comment 1-2

In case your model considers conduction, please clarify to which extend the ground is modelled (area, material, etc.)!

The photogrammetrically generated surface mesh inherently contains the interface between the body and the ground, as both material classes can be segmented within the 3D model. After translation of this surface mesh into a 3D cubic grid, the ground is thus modelled by assigning the thermal conductivity of the ground material to the appropriate cubes within the grid. For the validation measurements in the morgue this corresponded to a stainless-steel medical dissection table with a thermal conductivity value of $14.35 \text{ W m}^{-1} \text{ K}^{-1}$. Moreover, in order to address concerns raised by all reviewers, we have applied our method at a variety of real (indoor and outdoor) crime scenes (see also the section titled 'General response to all reviewers' above). In this newly added field validation study, we encountered a variety of ground materials such as concrete, wooden floors and a mattress, for which

we assigned thermal conductivities of $1.7 \text{ Wm}^{-1}\text{K}^{-1}$, $0.07 \text{ Wm}^{-1}\text{K}^{-1}$, and $0.05 \text{ Wm}^{-1}\text{K}^{-1}$, respectively. We have added this information to the manuscript (see lines 406-408).

Comment 1-3

In case your model considers convection, please name the parameters that are taken into account!

The supplementary material section of our previous paper contains a detailed discussion of our convection model (Sci. Adv. 6, eaba4243 (2020), doi:10.1126/sciadv.aba4243). Briefly, we model the convective energy transfer between cubes as $\Delta E = h_{conv}A\Delta T\Delta t$ where $h_{conv} = \frac{Nu \lambda}{L}$ with Nu being the Nusselt number for combined free and forced convection computed via $(Nu_{free}^3 + Nu_{forced}^3)^{\frac{1}{3}}$. Free convection is modelled as: $Nu_{free} = 0.63Gr^{0.25}Pr^{0.25}$, where Gr and Pr denote the Grasshof number for a vertical flat plate and the Prandtl number, respectively. We model forced convection both for the laminar and turbulent flow regimes. For the laminar flow regime, we compute Nu_{forced} by averaging the Nusselt number for flow along a plane and a cylinder (as an approximation of the human body) using:

$$Nu = \left(0.3 + \frac{0.62Re^{0.5}Pr^{\frac{1}{3}}}{\left(1 + \left(\frac{0.4}{Pr}\right)^{\frac{2}{3}}\right)^{0.25}} \cdot \left(1 + \left(\frac{Re}{2.82 \cdot 10^5}\right)^{0.625}\right)^{0.8} + \frac{0.6774Re^{0.5}Pr^{\frac{1}{3}}}{\left(1 + \left(\frac{0.0468}{Pr}\right)^{\frac{2}{3}}\right)^{0.25}} \right)$$

In the turbulent regime, we calculate Nu_{forced} using:

$$Nu = \frac{0.6774Re_{crit}^{0.5}Pr^{\frac{1}{3}}}{\left(1 + \left(\frac{0.0468}{Pr}\right)^{\frac{2}{3}}\right)^{0.25}} + 0.037 Pr^{\frac{1}{3}} (Re^{0.8} - Re_{crit}^{0.8})$$

Here, Re and Re_{crit} denote the Reynolds and the critical Reynolds (in our case $Re_{crit} = 5 \cdot 10^5$) number, respectively. Since this aspect of our model has been published previously, we believe it is not necessary to describe it in detail in our current manuscript. However, we now make it clearer in the manuscript that this information can be found in our previous publication (see line 419).

Comment 1-4

In case your model considers the radiation transfer, please clarify to which extend is the surrounding environment modelled?

Our model of the radiative heat transfer is also detailed in the supplementary material section of our previous paper (Sci. Adv. 6, eaba4243 (2020), doi:10.1126/sciadv.aba4243). In short, our TFD algorithm models the cube-wise radiative energy transfer as $\Delta E = h_{rad}A\Delta T\Delta t$ where $h_{rad} = \sigma\epsilon(T_b^2 + T_\infty^2)(T_b + T_\infty)$. Here, σ , ϵ , T_b and T_∞ denote the Stefan-Boltzmann constant, the material emissivity, the body temperature and the ambient temperature, respectively. We derive this description directly from the Stefan-Boltzmann equation for radiative energy transfer, under the assumption that the difference between T_b and T_∞ is small. Radiation is therefore mainly included as a heat transfer pathway *away* from the body *into* the environment. We now make it clearer in the manuscript that this information can be found in our previous publication (see line 421).

Comment 2

Please, give a short remark how do you determine (calculate) the best approximation of two spatial temperature distributions (measured/simulated)!

In this study, we reconstruct the PMI of a given body in two steps. First, we simulate the temporal evolution of the bodies' spatial temperature distribution over a given period of time (e.g., 48h). Second, we find the time point within this simulation at which the simulated temperature *at a specific body location* (which corresponds to a *single* cube in the individualised TFD model) best resembles its *co-localised* (through the use of imaging targets) measured temperature. This corresponds to finding the time point at which the *difference* between the *measured* and the *simulated location-specific* temperature is *minimal/smallest*. In this way we independently reconstruct PMIs based on the simulated and measured data from five distinct measurement locations (cf. Fig. 1). We have amended the manuscript to state this more clearly (see lines 128-129).

Comment 3

Please, explain why you first generating a triangle surface mesh. The point cloud from SfM can directly transferred into a volumetric representation (voxels).

By generating a surface mesh as the topological boundary of the point cloud, we gain access to its topological properties, such as the outer-pointing normal. This, in turn, allows us to accurately model postures resulting in non-convex hulls of the point cloud (*i.e.*, body postures which result only in partial contact with the substrate, such as bent limbs, arched small of the back *etc.*) by providing a definition of “inside/outside” of the point cloud. Moreover, this representation reduces spatial noise, reduces file size and facilitates memory management. We have added this information to the manuscript (see lines 396-402).

Comment 4-1

Please, clarify for the practical (operational) use of the proposed method: You are considering skin temperatures, right? Has the entire clothing to be removed before thermography?

In light of concerns regarding the practicality of our method raised by all reviewers, we carried out a unique field validation study in collaboration with the Dutch National Police and the Netherlands Forensic Institute (see also the section titled ‘*General response to all reviewers*’ above). The results of this field validation have been added to the revised manuscript. Here, we applied our method at a variety of real indoor and outdoor crime scenes, most of which involved clothed bodies. As a result, small incisions were made in the clothing (*i.e.*, without undressing the body) through which the thermal loggers were attached to the skin. The same approach would be suitable for thermography measurements.

Comment 4-2

If YES, how do you prevent contamination and destruction of other traces?

To prevent contamination and destruction of other traces in our field validation study, we applied our method only *after* all other trace sampling was completed. Additionally, the thermal loggers were sterilised/decontaminated before each use. This information is included in the revised manuscript (see lines 446-449). Moreover, both implementations of our method are in the very least *non-invasive* as they utilise *external* skin temperatures. Consequently, our method reduces contamination risk compared with the invasive rectal measurements necessary for use of the gold standard (and all actions required to gain access to the measurement location).

Comment 4-3

If YES, Is this the required time included in the 10min?

Our field validation study at real crime scenes revealed that the total time including incising of the clothing on average did not exceed ~15min.

Comment 5-1

Please, address briefly the challenges and limitations of the approach at the end of the paper and distinguish between indoor and outdoor simulations.

We have added a discussion of the challenges and limitations in both indoor and outdoor settings to the discussion section of the manuscript (see lines 298-317).

Comment 5-2

Please, address for outdoor simulation the problem of parameter setting for the post-mortem interval with unknown starting situation (time of death). During the PMI the parameter set of the outdoor environment typically changes over time, e.g., in the morning: increasing radiation by sun, ground temperature, temperature of nearby objects, wind, humidity, rain, etc.)

We agree with the reviewer that unknown changes in the environmental parameters increase the uncertainty in the reconstructed PMI. However, in contrast with the gold standard - and due to the time-resolved nature of our simulation algorithm - our approach allows modelling, and hence PMI determination, for a variety of scenarios (such as the changing ambient conditions mentioned by the reviewer). Moreover, our method is compatible with optimisation strategies where unknown model parameters are varied until the difference between the location-specific measured and simulated temperature evolutions (cooling curves) is minimal. We have added this information to the manuscript (see lines 298-309).

Comment 6

Could you give a statement based on the comparison of both approaches concerning their potential for further investigations? Is it an advantage that thermography provides the temperature distribution over the entire hull of the body instead of just measuring single points by temperature loggers?

The thermographic implementation provides, as pointed out by the reviewer, a much larger number of separate measurements. These, in turn, can be used to derive more densely sampled PMI distributions. As a result, the outcomes of the thermographic implementation (if combined appropriately) could yield more robust PMI determinations. In addition, as thermal images retain spatial information, they contain more case-specific information, which may be useful at a later stage of the investigation. At the same time, the currently much higher procurement costs for the necessary hardware may hamper widespread adoption of the thermographic approach in the forensic field. Likewise, utilising the unreduced thermographic data set will result in increased computation times, potentially limiting acceptance by crime scene investigators. In contrast, the use of the less costly thermal loggers is simple and requires no additional training, in turn likely facilitating rapid adoption by forensic practitioners. Technological strides may, however, improve the speed and affordability of the full-body thermographic approach in the future. We have revised the manuscript to include this information (see lines 318-331).

Comment 7

Please, could name the point density (pts/cm³) of the point cloud which was received from the adapted RGB (464 x 348 pixel) of the FLIR camera?

In this study, we did not generate point clouds from the FLIR data. We instead generated the point clouds from the DSLR data. The average point densities (\pm standard deviation) for the morgue and crime scene point clouds were 32 ± 51 pts/cm³ and 31 ± 27 pts/cm³, respectively. We have added this information to the manuscript (see lines 392-395).

Response to Reviewer #3

The authors have tapped into a unique set of procedures for estimating the PMI of human remains. As pointed out in their manuscript, determining the PMI accurately and precisely remains a massive challenge for the forensic sciences community. I found the manuscript to be well-written, concise, and thought provoking. However, I also recognized several limitations to the study that the authors should address. The two major limitations of the current study are, 1) lack of sample size from their location or from others around the world, and 2) no validation.

We agree with the reviewer's first point that it is desirable to have as large a sample size as possible. Given that our morgue-based validation is performed on human remains, obtaining very high sample sizes is, naturally, limited. Nonetheless, we have strived to obtain the highest possible sample size; and we have increased our sample size compared with our previously published study (Sci. Adv. 6, eaba4243 (2020), doi:10.1126/sciadv.aba4243). We therefore believe that our sample size is sufficiently large to validate our method. Regarding the suggestion to use data from others around the world: as our approach represents the first of its kind, the required co-registered data currently do not exist in any international databases. Furthermore, given the generality of our approach, applicability can reasonably be assumed to be global.

With regard to the reviewer's second point, we do not agree that no validation was performed; our method was validated on recently deceased human bodies which were transported to our hospital morgue as part of our scientific body donation programme. However, we acknowledge that these morgue-based validation studies may not necessarily imply accurate PMI determination at real crime scenes. To alleviate this potential concern, and also in light of comments from reviewers 1 and 2, we have subsequently carried out a unique field validation study in collaboration with the Dutch National Police and the Netherlands Forensic Institute (see also the section titled '*General response to all reviewers*' above). As well as increasing our sample size, this field validation renders our study the first non-subjective, individualised and non-contact thermometric PMI reconstruction method to be validated in forensic practice. Consequently, our manuscript now comprises results from *two separate validation studies*: (i) *lab validation* in our hospital morgue and (ii) *field validation* at various real indoor and outdoor crime scenes. We believe that these improvements and clarifications address the concerns raised by the reviewer regarding both our sample size and validation procedure.

REVIEWER COMMENTS

Reviewer #1 (Remarks to the Author):

The authors replied to all questions and remarks raised by the reviewers appropriately. I therefore suggest acceptance of the article.

Reviewer #2 (Remarks to the Author):

My remarks were addressed in the rebuttal and mostly considered in the revised version of the paper. It remains only a minor remark concerning "Comment 4-1"

R: Please, clarify for the practical (operational) use of the proposed method: You are considering skin temperatures, right? Has the entire clothing to be removed before thermography?

Authors answer:

A: In light of concerns regarding the practicality of our method ... As a result, small incisions were made in the clothing (i.e., without undressing the body) through which the thermal loggers were attached to the skin. The same approach would be suitable for thermography measurements.

Authors text:

448 sterilised/decontaminated before each use. In cases where the body was clothed, small incisions
449 were made in the fabric (without moving or undressing the body) to ensure contact of the thermal
450 loggers with the skin.

Please, consider thermography of clothed bodies (e.g., in line 450)

450 loggers with the skin or mapping by thermography.

Reviewer #3 (Remarks to the Author):

First, and foremost, I appreciate the authors expansion of their validation. But, I still find the validation ($n = 4$ with 2 in one environment and 2 in another; take into account the number per sex and degrees of freedom are further reduced) to be extremely weak given the conclusions drawn. At minimum, you should state such limitations. For example, the proposed method does not take into account season or location; thus, your work is highly limited.

Furthermore, to suggest there is a gold standard (i.e., rectal temperature) when such a concept in itself is sorely misleading (i.e., you indicate that such methods exist- even DNA, which is considered a 'gold standard' alone has massive issues with observer bias) lacks responsibility. As a practicing basic and applied researcher and forensic practitioner, I recognize individuals in court seize on such statements and, unfortunately, misuse it in courts of law.

For me to recommend acceptance, the authors would need to be extremely clear about the limitations of their work and what needs to be done.

Response to reviewers' comments

Below we respond in detail to each of the reviewers' comments and describe how we have adapted our manuscript accordingly. All changes are highlighted in red text in the revised manuscript.

Response to Reviewer #1

The authors replied to all questions and remarks raised by the reviewers appropriately. I therefore suggest acceptance of the article.

We are happy that the reviewer is satisfied with our changes and we thank the reviewer for their earlier comments which we feel greatly improved our manuscript.

Response to Reviewer #2

It remains only a minor remark concerning "Comment 4-1": Please, consider thermography of clothed bodies (e.g. in line 450).

We thank the reviewer for this useful suggestion and have adapted the manuscript as follows (line 458-462):

All thermal loggers were sterilised/decontaminated before each use. In cases where the body was clothed, small incisions were made in the fabric (without moving or undressing the body) to ensure contact of the thermal loggers with the skin (or exposure of the skin for non-contact thermometry using a thermal camera).

Response to Reviewer #3

First, and foremost, I appreciate the authors expansion of their validation. But, I still find the validation ($n = 4$ with 2 in one environment and 2 in another; take into account the number per sex and degrees of freedom are further reduced) to be extremely weak given the conclusions drawn. At minimum, you should state such limitations. For example, the proposed method does not take into account season or location; thus, your work is highly limited.

Furthermore, to suggest there is a gold standard (i.e., rectal temperature) when such a concept in itself is sorely misleading (i.e., you indicate that such methods exist- even DNA, which is considered a 'gold standard' alone has massive issues with observer bias) lacks responsibility. As a practicing basic and applied researcher and forensic practitioner, I recognize individuals in court seize on such statements and, unfortunately, misuse it in courts of law.

For me to recommend acceptance, the authors would need to be extremely clear about the limitations of their work and what needs to be done.

We acknowledge the points raised by the reviewer. Although we demonstrated that our approach has a high degree of accuracy in both lab-based validation (N=6) and field validation (N=4) measurements, we fully accept that the total number of body samples is limited, and that we do not explore the effects of variations such as changes in season and geographical location. We agree that these limitations should be stated more clearly. To address this, we have added a new paragraph to our discussion section, where we state that further validation of the technique in other environmental conditions as well as a large number of comparative validations are necessary to establish the method's full range of applicability and potential future permissibility in forensic practice (see lines 282-289).

Furthermore, we accept the reviewer's concern regarding the term 'gold standard' and have therefore replaced all instances in the revised manuscript with the term 'standard method'.

Response to Reviewer #4

Reviewer #4 reiterated that the standard method used here as comparison is based on hundreds of validation experiments, and thus the comparison and the validation should be performed in as many experiments before being adopted, and this would take years before claiming that indeed this method is superior.

We agree with the reviewer that in order to unequivocally establish superior accuracy as well as potential future permissibility of our approach, the number of comparative validation studies carried out should at least equal those underlying the standard method. We have added this to the manuscript (see lines 286-289).

REVIEWERS' COMMENTS

Reviewer #3 (Remarks to the Author):

The authors have addressed my concerns- nicely done study!

Response to reviewers' comments

Below we respond to the reviewer's comments.

Response to Reviewer #3

The authors have addressed my concerns- nicely done study!

We thank the reviewer for their helpful suggestions which we feel greatly improved our manuscript.